# Adaptive Articulation Angle Preview-Based Path-Following Algorithm for Tractor-Semitrailer Using Optimal Control

**DOI:** 10.3390/s22145163

**Published:** 2022-07-10

**Authors:** Xuequan Tang, Yunbing Yan, Baohua Wang, Lin Zhang

**Affiliations:** 1Department of Automobile and Traffic Engineering, Wuhan University of Science and Technology, No. 2, Huangjiahu West Road, Hongshan District, Wuhan 430065, China; txuequantang@wust.edu.cn (X.T.); zhanglin4025@wust.edu.cn (L.Z.); 2Department of Automobile Engineering, Hubei University of Automotive Technology, Shiyan 442002, China; 19950009@huat.edu.cn

**Keywords:** articulated vehicles (AVs), Path-Following Algorithm, closed-loop simulation, preview, optimal control

## Abstract

Most existing Path-Following Algorithms (PFAs) are developed for single-unit vehicles (SUVs) and rarely for articulated vehicles (AVs). Since these PFAs ignore the motion of the trailer, they may cause large tracking deviations and ride stability issues when cornering. To this end, an Adaptive Articulation Angle Preview-based Path-Following Algorithm (AAAP-PFA) is proposed for AVs. Different from previous PFAs, in this model, a simple linear vehicle dynamics model is used as the prediction model, and an offset distance calculated by an articulation angle is used as part of the preview distance. An adaptive posture control strategy is designed to trade off the trajectory tracking performance and lateral stability performance during the path-following process. Considering a large prediction mismatch caused by using a linear vehicle dynamics model, a feedback correction method is proposed to improve the robustness of the steering control. In the comparison simulation experiment with SUV-PFA, it is confirmed that the novel PFA has better adaptability to the contradictory relationship between tracking performance and lateral stability and has strong steering control robustness.

## 1. Introduction

As an important part of the human–vehicle–road closed-loop system, the driver is responsible for sensing the road environment and vehicle motion state and controlling the direction and speed of the vehicle [1,2]. Developing advanced driver models can reduce the cost of real driving experiments and avoid the danger of experimenters. In addition, advanced driver assistance systems and active safety systems can be developed through these closed-loop experiments [3,4]. Path following is a vital function of the driver model. Some active steering control systems use preview Path-Following Algorithms (PFAs) to actively intervene in wheel steering input and drive/braking torque to improve driving stability [5,6,7,8,9]. Different from the active safety control system, PFAs focus on the steering control ability during cornering, which is reflected in the stability and accuracy of steering control.

To date, a variety of PFAs has been developed. The Optimal Preview Control Path Driver model (OPCDM) proposed by MacAdam is the most representative [10,11]. The model is still used in commercial software such as CarSim. With the development and maturity of intelligent control theory, Neural Network-based Path-Following Algorithm [12,13,14], Fuzzy Control-based Path-Following Algorithm [15,16], Model Prediction Control-based Path-Following Algorithm [17,18], Machine Learning-based Path-Following Algorithm [19,20] and so on, have been proposed. These models are characterized by more predictive information, stronger robustness, more complex structure design and a higher degree of personification. However, most of these PFAs were applied to single-unit vehicles (SUVs).

Some PFAs suitable for articulated vehicles are investigated. Zhaoheng Liu proposed a method to study the lateral and yaw dynamics of the semitrailer based on the optimal Path-Following Algorithm [21]. State feedback with delay was added to the model, and the model parameters were identified by the Quasi-Newton optimization method. Li Xiansheng added the real-time motion state information of the semitrailer into the calculation of vehicle steering control, proposed a semitrailer Path-Following Algorithm based on the Quasi-Newton optimization method and studied the influence of semitrailer load, speed and a driver’s inherent time delay on vehicle behavior [22]. In order to improve the lateral stability of articulated vehicles (AVs), Yuping He proposed the Lateral Position Deviation Preview (LPDP) model based on LQR [23]. This model was composed of a tractor tracker and a trailer tracker. The lateral displacement of each axle was previewed, and the steering wheel angle of the front car and the rear car was deduced. The cost function of lateral acceleration was designed to solve the optimal steering wheel angle. Shenjin Zhu proposed the Unified Lateral Preview Driver Model, which uses the linear 4-DOF dynamics model of AVs as a reference model, and the lateral displacement deviation of each unit were predicted for a certain time through the transfer matrix [24].

The differences from SUVs are that AVs are structurally articulated with one or more trailers, with a longer body, and have a higher center of mass when loading, larger turning clearance circle and obvious nonlinear dynamics characteristics when turning [25,26]. The shortcomings of the PFAs reviewed above are as follows:(1).For SUV-PFA, obvious deviation occurs when following the trajectory at low speed, and lateral stability will be reduced when performing evasive maneuvers at high speeds [24].(2).AVs exhibit obvious nonlinear characteristics and many uncertain factors during maneuvering. The PFAs established by the linear vehicle dynamics model is difficult to accurately predict the motion state of all units, which will reduce the robustness of steering control.(3).Ignoring the motion state of the trailer, it is difficult to avoid dangerous situations such as sideslip and rollover. Few PAFs consider the influence of the motion posture and steering kinematics of AVs to improve performance. Maintaining a good posture during maneuvers can effectively improve lateral stability [27].

Aiming at the above shortcomings, an Adaptive Articulation Angle Preview-based Path-Following Algorithm (AAAP-PFA) considering the steering kinematics and driving attitude of the semitrailer is proposed. The following are the paper’s main contributions: (1) By analyzing the change of turning clearance circle and posture of AVs body units when cornering, as part of the preview information, the articulation angle is used to calculate the total preview deviation; (2) In order to solve prediction mismatch, a Feedback Correction (FC) method is proposed to improve the robustness of the steering control; and (3) an adaptive posture control strategy that considers sideslip, swing and roll of all body units is designed to trade off trajectory tracking and lateral stability performance. In order to verify the superiority of the proposed PFA, a closed-loop simulation experiment was designed to evaluate and explain the performance.

## 2. Vehicle Models

The three DOF linear yaw-dynamics model of a tractor-semitrailer is derived as the predictive model to design the novel PFA, as shown in Figure 1. The predictive model needs to be able to accurately describe the dynamic response of the vehicle but also to be as simplified as possible to improve the execution efficiency of the control algorithm. O_1_ and O_2_ are the centroids of the tractor and semitrailer, respectively. *θ* is the articulation angle. *F_yf_*, *F_yr_*, *F_yt_* and *F_a_* are the total cornering force of the front and rear axle tire of the tractor, the total rear axle tire of the semitrailer and the lateral force of the hitch, respectively. *Ψ* is the yaw angle of the tractor in the global coordinate system. The three degrees of freedom are: yaw movement (*r*_1_) and lateral movement (*v_y_*) of the tractor, and the articulation angle (*θ*). *δ_f_* is the wheel angle of the front axle.

With reference to the modeling methods in previous studies [23,28] and considering the necessary dynamic response characteristics, the multi-axle system is simplified to a single-axle system. The following assumptions are made: (1) forward speed is a constant value and is guaranteed by a PID algorithm embedded in the software; (2) dynamics of the tire are ignored, and the linear tire model is adopted; (3) the articulation angle and the tire slip angle are small, and sin *δ_f_* ≈ *δ_f_*, sin *θ* ≈ *θ*, cos *δ_f_* ≈ 1, cos *θ* ≈ 1; and (4) ignore the influence of steering system, suspension system, etc., vehicle parameters remain unchanged. Although linear tires can reduce the accuracy of the prediction model at high speeds, they can improve the calculation speed and reduce the complexity. In the following chapters, the FC method can effectively make up for the shortage of decreased prediction accuracy. The symbols and physical meanings of vehicle dynamics parameters are shown in Table 1.

According to the dynamics theory of the vehicle system and Newtonian mechanics principle, the dynamic equations of the tractor are as follows:(1)$$m_{1}\left({\dot{v}}_{y}+v_{x}r_{1}\right)=F_{yf}+F_{yr}-F_{a}$$
(2)$$I_{Z1}{\dot{r}}_{1}=F_{yf}a_{1}-F_{yr}b_{1}+Yc_{t}$$

The dynamic equations of the semitrailer are as follows:(3)$$m_{2}\left({\dot{v}}_{y}+v_{x}r_{1}-c_{t}{\dot{r}}_{1}-a_{2}{\dot{r}}_{2}\right)=F_{a}+F_{yt}$$
(4)$$I_{Z2}{\dot{r}}_{2}=F_{a}a_{2}-F_{yt}b_{2}$$

The change of the articulation angle can be expressed as:(5)$$\dot{\theta }=r_{1}-r_{2}$$

In the global coordinate system, the lateral motion and yaw angle change equation of the tractor is:(6)$$\left\{\begin{array}{l} {\dot{Y}}_{V}^{G}=v_{x}\sin(\psi )+v_{y}\cos(\psi )\\ \;\;\dot{\psi }=r_{1} \end{array}\right.$$

The cornering force on the front and rear axles of the tractor and the rear axle of the semitrailer are, respectively, expressed as:(7)$$\left\{\begin{array}{l} F_{yf}=-C_{yf}\alpha_{f}=-C_{yf}(\frac{v_{y}+a_{1}r_{1}}{v_{x}}-\delta_{f})\\ F_{yr}=-C_{yr}\alpha_{r}=-C_{yr}\frac{v_{y}-b_{1}r_{1}}{v_{x}}\\ F_{yt}=-C_{yt}\alpha_{t}=-C_{yt}(\frac{v_{y}-c_{t}r_{1}-L_{2}r_{2}}{v_{x}}+\theta ) \end{array}\right.$$

The above equation can be expressed as a state-space expression model as follows:(8)$$\left\{\begin{array}{l} \dot{X}=AX+BU\\ \;\;Y=CX \end{array}\right.$$
where Matrices *A* and *B* are provided in the Appendix A. *X* is a state vector. To predict the lateral displacement and yaw angle, define $Y={\left[\begin{array}{cc} Y_{P^{\prime }}^{V} & \psi_{P^{\prime }}^{V} \end{array}\right]}^{T}$. *C* is an output vector given in Section 3.2 and $U=\left[\delta_{f}\right]$ is the input.

## 3. Adaptive Articulation Angle Preview-Based Path-Following Algorithm

Figure 2 is the overall block diagram of AAAP-PFA. The construction idea of the proposed FPA is as follows: the desired trajectory is used as the input, the feedback channel is formed by the predicted vehicle trajectory, the FC for model prediction mismatch error and the output of the adaptive posture control strategy and the optimal steer angle is solved according to the difference between the input and the feedback.

In order to implement the algorithm, the steps are as follows:(1)The turning characteristics of the tractor-semitrailer are analyzed in Section 3.1;(2)The calculation method of the motion state of all vehicle units and the FC method are derived in Section 3.2;(3)The posture control strategies of articulated vehicle units are designed in Section 3.3;(4)Based on the optimal control theory, the optimal steering angle of the novel PFA is derived in Section 3.4, and the stability analysis is carried out in Section 3.5.

### 3.1. Turning Kinematics Characteristics of Tractor-Semitrailer

Figure 3 shows a schematic diagram of the turning kinematics of a tractor-semitrailer. The circular area swept by the trajectories of the outer wheels of the front axle of the tractor and the inner wheels of the rear axle of the semitrailer is the turning channel circle. Point A is the intersection of the front axis of the tractor and the center line of the Turning Channel Circular (TCC), point B is the center of the front axle of the tractor, and point O is the instantaneous center of rotation. When a tractor-semitrailer turns, the sweep area of TCC will become larger. When turning in a steady-state, the instantaneous turning center of the tractor and the trailer coincide, the turning radius of the rear-inner wheel of the trailer is the minimum, and the turning radius of the front-outer wheel of the tractor is the maximum.

In a sense, the center line of the TCC represents the trajectory of the tractor-semitrailer. As part of the total preview lateral displacement deviation, the offset distance can make the center line as close as possible to the desired trajectory. However, it is difficult and complicated to calculate the offset distance accurately. For simplicity, the offset distance AB is approximated by the distance between the semitrailer centroid deviating from the longitudinal axis of the tractor and is defined by:(9)$$y_{offset}(t)=k_{offset}a_{2}\theta (t)$$
where *k_offset_* is the proportional coefficient and is named the posture compensation coefficient. The offset distance is used to control the posture of vehicle units and the trajectory of TCC.

### 3.2. FC Method and Vehicle State Prediction

Predicting the vehicle state for a period of time in the future is the basis for steering control, and the preview principle of AAAP-PFA is shown in Figure 4. The preview starting point is fixed at the center of the front axle of the tractor. Point P is the preview point. *T_P_* is the time required for the vehicle to travel to point P’ from the preview starting point. In engineering practice, the real-time longitude and latitude of the vehicle in the geodetic coordinate system and the preview point in the planned path can be calculated by positioning and navigation units [29,30]. The real-time motion state of the vehicle can be estimated more accurately by using Kalman filtering and other fitting algorithms [31]. The driver estimates the vehicle state and its relative position to the target trajectory by feeling [32,33].

(1) Coordinate transformation. The ordinate of the preview point P and point P’ in the vehicle coordinate system can be obtained as:
(10)$$\left\{\begin{array}{c} Y_{{}_{P}}^{V}=[Y_{{}_{P}}^{G}-Y_{{}_{V}}^{G}(t)]\cos\psi -[X_{{}_{P}}^{G}-X_{{}_{V}}^{G}(t)]\sin\psi \\ Y_{{}^{P^{\prime }}}^{V}(t+T_{P})=[Y_{{}_{V}}^{G}(t+T_{P})-Y_{{}_{V}}^{G}(t)]\cos\psi -[X_{{}_{V}}^{G}(t+T_{P})-X_{{}_{V}}^{G}(t)]\sin\psi \end{array}\right.$$

Note that the superscript and subscript of the symbols in this section are the coordinate system and the object (vehicle body, desired trajectory and points), respectively. The abscissa of the preview point P in the vehicle coordinate system is the product of *T_PT_* and the vehicle speed *v_x_.*

(2) Vehicle state and trajectory prediction. Using the state transition matrix to predict the states and output of the vehicle units at time *t +* Δ*t* is:
(11)$$\left\{\begin{array}{l} X(t+\Delta t)=e^{A\Delta t}X(t)+\int_{0}^{\Delta t}e^{A\eta }d\eta BU\\ Y(t+\Delta t)=Ce^{A\Delta t}X(t)+C\int_{0}^{\Delta t}e^{A\eta }d\eta BU \end{array}\right.$$

Let us define matrix *G* and vector *F* as:(12)$$\left\{\begin{array}{l} F=Ce^{A\cdot \Delta t}\\ G=C\int_{0}^{\Delta t}e^{A\cdot \eta }d\eta B \end{array}\right.$$

The above continuous prediction process is transformed into a discrete prediction process. Divide the time period from time t to *t + T_P_* into *N_P_* time intervals, then the time step is:(13)$$\Delta t=\frac{T_{P}}{N_{P}}$$

Through iterative calculation, we predict the output at each moment:(14)$$Y(t+k\Delta t|t)=FX(t+(k-1)\Delta t|t)+GU,\;k=1,2,3\cdots \frac{N_{P}}{\Delta t}$$

We define the output vector *C* as:(15)$$C=\left[\begin{array}{cccccc} 1 & a_{1} & 0 & 0 & 0 & 0\\ 0 & 1 & 0 & 0 & 0 & 0 \end{array}\right]$$

The output at time *t + T_P_* is:(16)$$Y(t+T_{P}|t)=F^{NP}X(t|t)+F^{NP-1}GU(t)+F^{NP-2}GU(t)\cdots +GU(t)$$

(3) FC for prediction mismatch. The parameter identification of the self-built vehicle model is not accurate enough; nonlinear characteristics, load transfer, vehicle speed changes and loading conditions and other uncertain factors will cause the robustness of the steering control of the PFA to deteriorate. Aimed at this problem, an FC method is adopted to compensate for the preview deviation.

At time *t − t*_1_, the vehicle’s state can be predicted at time *t* as:(17)$$Y^{V}(t|t-t_{1})=F^{\frac{t_{1}}{\Delta t}}X(t-t_{1}|t-t_{1})+F^{\frac{t_{1}}{\Delta t}-1}GU(t-t_{1})+F^{\frac{t_{1}}{\Delta t}-2}GU(t-t_{1})\cdots +GU(t-t_{1})$$

Considering the calculation efficiency and the accumulated error in the prediction process, after many simulation tests, when 0.02 *T_P_* ≤ *t*_1_ and Δ*t* ≤ 0.2 *T_P_*, a better control effect can be achieved.

The prediction deviation at time *t* is:(18)$$e_{prediction}(t)=Y^{V}(t|t-t_{1})-Y^{V}(t)$$

The compensation deviation of FC is:(19)$$y_{fc}(t)=\left[\begin{array}{cc} k_{fc1} & k_{fc2} \end{array}\right]e_{prediction}(t)$$
where *k_fc_*_1_ and *k_fc_*_2_ are the weight coefficients of the lateral displacement prediction deviation and the yaw angle prediction deviation, respectively. These two values need to be adaptively adjusted according to the accuracy of the prediction results.

The total lateral displacement preview deviation and yaw angle preview deviation is:(20)$$\left\{\begin{array}{l} e_{Y}(t)=Y_{P}^{V}-Y_{P^{\prime }}^{V}(t+T_{P})-y_{offset}(t)-y_{fc}(t)\\ \;e_{\psi }=\psi_{P}^{V}-\psi_{P\prime }^{V}(t+T_{P}) \end{array}\right.$$

### 3.3. Adaptive Posture Control Strategy

In directional control, the posture of all vehicle units determines the trajectory of the TCC centerline and the motion state, which directly affect the trajectory tracking performance and lateral stability performance. The basis of the adaptive attitude control strategy design is how to adjust the posture to respond to a change in the desired trajectory. Figure 5 shows the principle of posture control. The vehicle units are simplified into an articulated axle system, and three possible postures are considered when turning to the left. The postures of the vehicle unit can be represented by the articulation angle. Figure 5b shows the control flow.

As shown in Figure 5a, when postures 1 and 3 are transformed into posture 2, the amplitudes of sideslip, roll and swing of the vehicle units will be reduced, and the lateral stability performance will be improved. During path following, the posture changes are the response to the changes in road curvature and lateral displacement. The increase in the articulation angle and the front wheel angle will make the lateral displacement deviation and the yaw angle deviation approach zero faster but increase the yaw rate and lateral acceleration of all vehicle units. As a result, the two performances are diametrically opposite and require a trade-off. The above rules also apply to the situation where the preview point is on the right side of the longitudinal axis of the tractor.

As shown in Figure 5b, the function of the posture control strategy is to adjust the offset distance in response to changes in articulation angle, roll, sideslip and swing of the body units to obtain the optimal steer angle. Let us define the preview point on the left side of the longitudinal axis of the tractor and the articulation angle rotating clockwise relative to the tractor as positive and the right as negative. Table 2 shows the effect of the sign of *k_offset_* on the preview deviation under three postures. “+” indicates an increase, and “−” indicates a decrease. The above rules also apply to the situation where the preview point is on the right side of the longitudinal axis of the tractor.

Roll, sideslip and swing are considered in the posture control strategy, and the evaluation index is defined as:(21)$$M=\sqrt{w_{1}{\left(\frac{\phi }{\phi_{0}}\right)}^{2}+w_{2}{\left(\frac{\beta }{\beta_{0}}\right)}^{2}+w_{3}{\left(\frac{\theta }{\theta_{0}}\right)}^{2}+w_{4}{\left(\frac{\dot{\theta }}{{\dot{\theta }}_{0}}\right)}^{2}}$$
where *M* ∈ [0, 1], *w*_1_, *w*_2_, *w*_3_ and *w*_4_ are weight coefficients, and *θ*_0_, *θ’*_0_, *β*_0_ and *ϕ*_0_ are reference values.

The adaptive control logic between the evaluation index *M* and *k_offset_* is:(22)$$k_{offset}=-M(pM-q)$$
where *p* and *q* are constants.

### 3.4. Optimal Steering Angle

Referring to the modeling process of OPCDM [10], an objective function that minimizes the sum of squares of preview deviations is established:(23)$$\min J=\frac{1}{N_{P}}\sum_{k=1}^{N_{P}}{\left\|\left[\begin{array}{c} Y_{P}^{V}(t+k\Delta t|t)-Y^{V}(t+k\Delta t|t)-y_{offset}(t|t)-y_{fc}(t+k\Delta t|t)\\ \psi_{P}^{V}-\psi_{P\prime }^{V}(t+T_{P}) \end{array}\right]\right\|}_{Q}^{2}w(k)$$

The meaning of the above formula is: in the discrete time period from time *t* to *t + T_P_*, the second norm of the lateral displacement deviation and the yaw angle deviation of the trajectory of the tractor in the vehicle coordinate system is the smallest. *Q* = [*q*_1_ 0; 0 *q*_2_] is the weight matrix of lateral displacement deviation and yaw angle deviation; *q*_1_ and *q*_2_ are the weight coefficients, respectively.

Solve the derivative of the objective function with respect to the input *U*, and make it zero:(24)$$\frac{\partial J}{\partial U}=0$$

There is a certain time delay between the calculated optimal steer angle and the current steer angle [16]; add a delay link:(25)$$H(s)=e^{-t_{d}s}$$

The optimal steering angle is:(26)$$\delta_{f}{\left(t+t_{d}\right)}^{*}=\delta_{f}(t)+\frac{\sum_{t_{k}=t+k\Delta t}^{t+T_{P}}{\left[\begin{array}{c} Y_{P}^{V}(t_{k}|t)-F_{1}X^{V}(t_{k}|t)-y_{offset}(t)-y_{fc}(t)\\ \psi_{P}^{V}(t_{k}|t)-F_{2}X^{V}(t_{k}|t) \end{array}\right]}^{T}QGw(k)}{\sum_{k=1}^{N_{P}}G^{T}QGw(k)}$$
where, *F*_1_ = *F*(1, :), *F*_2_ = *F*(2, :).

### 3.5. Stability Analysis

In Equation (26), the numerator of the second term on the right side of the function represents the prediction result, and the denominator represents the forced response coefficient of the prediction link. The FC link can ensure the accuracy and convergence of the prediction results. Therefore, the stability of the PFA is completely determined by the matrix *G*. If the state-space system of the linear vehicle model is stable, then *T_P_* will be the only variable that affects the finiteness and convergence of matrix *G*. Set *g*_1_ = *G*(1, 1), and *g*_2_ = *G*(1, 2). The denominator can be expressed as:(27)$$G^{T}QG=q_{1}g_{1}^{2}+q_{2}g_{2}^{2}$$

Figure 6 shows the values of *g*_1_ and *g*_2_ under different preview times *T_P_*. *g*_1_ is more sensitive to increments of control input. When *T_P_* increases, the steering angle increment becomes smaller; that is, the gain of the lateral displacement deviation to the steering angle becomes smaller. When the preview time approaches zero, the increment of the control input approaches infinity, and the steering control of PFA becomes unstable. Hence, an appropriate size of *T_P_* is the key to the stability of the PFA controller and adjusting the weight coefficient can change the sensitivity of the directional bias and lateral displacement bias to the angle input.

## 4. Simulation

In TruckSim software, a nonlinear vehicle model of a 3A Tractor + 2A Semitrailer is established as the control object for the closed-loop simulation. The following performances of the proposed novel FPA will be verified: (1) Adjustment effect of articulation preview on trajectory tracking performance and lateral stability performance; (2) Robustness of steering control against prediction mismatch; and (3) The effectiveness of the adaptive posture control strategy.

### 4.1. Test Conditions and Parameter Settings

The extent to which the trajectories of the vehicle units are close to the desired trajectory, the size of the overshoot and the rapid response of direction can intuitively evaluate the performance of path following. The response curves for lateral acceleration and yaw rate are often used to assess the lateral stability performance. The common parameter settings are as follows: *Q* = [5 0; 0 1], *T_P_* = 1.0 s, *k_fc_*_1_ = *k_fc_*_2_ = 1, Δ*t* = 0.1 s, *w*(k) = 1.

(1) Test 1: Aimed at the first performance. A single lane change trajectory was used as the input to the trajectory tracking control system, with initial speeds of 80 km/h. The road adhesion coefficient is 0.8. OPCDM was used as a comparison object.

(2) Test 2: Different from test 1, the initial speeds are set to 40 and 80 km/h. By comparing the steady-state values of the nonlinear vehicle dynamic model established in TruckSim software with that of the self-built vehicle model under angular step conditions, the prediction accuracy was defined:
(28)$$Accuracy=\left|\frac{V_{Truck\mathrm{si}m}-V_{Self-built}}{V_{Truck\mathrm{si}m}}\right|$$

(3) Test 3: To simulate an environment with large sideslip, large roll and frequent swing, a sine trajectory is designed, as shown in Figure 7. The maximum lateral offset is *H* = 6 m, and the width is *B* = 50 m. The curve is constructed as follows:
(29)Y=0,             X≤x0H2−H2cos[2πB(X−x0)], x0<X≤x0+B0,           x0+B<X

The maximum curvature is:(30)$$K_{\max}=\frac{2H\pi^{2}}{B^{2}}$$

In addition, the road adhesion coefficient is 0.4, the initial speed is set to 80 km/h. The other parameter settings are as follows: *w*_1_ = *w*_2_ = *w*_3_ = *w*_4_ =1, *p* = 30, *q* = 12, *θ*_0_ =7.5, *θ*′_0_ = 6.5, *β*_0_ = 2, *ϕ*_0_ = 0.2.

### 4.2. Test 1

Figure 8 shows the path-following performance of AAAP-PFA at a speed of 80 km/h. From the tracking deviation at the overshoot (X = 100 m) and the yaw angle, it can be seen that the closeness of the trajectory of the vehicle units in the second straight line segment, the second curve and the end straight line segment to the target trajectory line is from small to large in the order of *k_offset_* = 0.6, OPCDM and *k_offset_* = −0.6. The state variables that characterize the lateral stability of the vehicle show the opposite laws, as shown in Figure 8e,f.

Table 3 shows the laws presented by the simulation results at bend 2. To better track the curvature changes of the desired trajectory, the PFA with a positive *k_offset_* can change the posture of the vehicle unit by increasing the steering angle, which will increase the lateral acceleration and yaw speed of the vehicle. This is detrimental to the lateral stability of articulated vehicles. On the contrary, the PFA with a negative *k_offset_* makes the trajectory of the body unit move away from the target trajectory. In this mode, the steering rate becomes slower, and the time for overshoot, yaw and articulation angles to reach a steady-state becomes longer, which means understeering. The slowly changing vehicle state is beneficial to the lateral stability of the vehicle.

For path following and evasive maneuvers at high speeds, in terms of importance, lateral stability performance is the first, followed by the ability not to deviate from the path. Therefore, the articulation preview method can be used to trend the PFA towards trajectory tracking performance or lateral stability performance. In summary, the parameter adaptive design of FPA is very valuable.

### 4.3. Test 2

Figure 9 shows the accuracy of steady-state value prediction and load transfer in the unit angle step experiment under different speeds and loading conditions. Without loading, the prediction accuracy of the self-built model is higher than 85%, except for the sideslip angle. When loaded, with the increase in speed, the prediction accuracy gradually decreases. Thus, load and speed are the key factors for inaccurate prediction. In addition, the increase in speed is the main inducement of the load transfer rate (LTR). Due to roll, the vertical load of the outer tire increases so that the nonlinearity of the tire increases. The self-built vehicle model will have a large deviation in predicting vehicle states. Therefore, the main factor that determines the model prediction results and the robustness of steering control is speed.

Figure 10 shows the simulation results of the AAAP-PFA with and without FC. It can be found that the trajectories of the two vehicle units with FC are closer to the desired trajectory under the two speeds, and the yaw angle error is smaller. Although there is a large overshoot at the end of the straight segment, the lateral displacement deviation and yaw angle approach zero significantly faster. In addition, the peak value of the steering angle, lateral acceleration and yaw rate are also increased. It can be inferred that the FC improves the robustness of the steering control.

### 4.4. Test 3

The comparison results of OPCDM and AAAP-PFA are shown in Figure 10. From Figure 11a, with the input of sine trajectory, the tracking process of the two control algorithms can be divided into five steps. There are two obvious differences, one of which is the second step, and the other is the third and fourth steps. The former corresponds to the point of maximum curvature, and the latter corresponds to transitions between the sine segment and straight segment. The results of the above two parts are described and analyzed as follows:

(1) Comparison of the first difference. From the lateral stability, the tractor-trailer with OPCDM shows a high sideslip at the point of maximum curvature, and AAAP-PFA, based on an adaptive posture control strategy, effectively limits the sideslip, as shown in Figure 11g. The reason is that the adaptive posture control strategy produces a large and negative offset, as shown in Figure 11e. This offset makes the steering wheel angle rapidly to the opposite direction and adaptively adjusts the posture of the vehicle units, as shown in Figure 11c,d. From Figure 11e, the process of posture adjustment is divided into two stages, namely: limit sideslip and fast approaching zero. The main reason why the novel PFA can prevent sideslip is that it can track the maximum curvature point and modify the posture and steering angle in real-time based on the vehicle states. In addition, the action of steering return causes a sudden yaw response and roll-jitter, as shown in Figure 11f,i. This phenomenon may bring unexpected consequences to the path-following process. Therefore, some measures of smoothing treatment need to be investigated.

(2) Comparison of the second difference. In this stage, in order to track the straight trajectory, the orientation and posture of the vehicle units must immediately reverse. As shown in Figure 11c,d, the tractor-trailer with AAAP-PFA undergoes six obvious adjustments in posture and orientation during the whole tracking process with a significant trend to zero. Further, the lateral displacement and yaw angle can approach zero quickly, and the magnitude and number of each adjustment are getting smaller and smaller, as shown in Figure 11a,b. Moreover, the yaw, swing and roll conditions of the vehicle units are gradually mitigated, as shown in Figure 11f,g,i. The reason for the above results is that a positive *k_offset_* is generated to bring the trajectories of the two body units closer to the desired trajectories, as shown in Figure 11e.

## 5. Conclusions

This paper proposes an AAAP-PFA applied for a tractor-semitrailer. The features of the original PFA are as follows: a 3-DOF linear yaw-model of tractor-semitrailer is used as the prediction model, the articulation angle as the preview information is adopted to affect the trajectory of the semitrailer, an FC link is added to solve the inaccurate prediction problem and an adaptive attitude strategy is proposed to adapt to the performance requirements of tracking and lateral stability during trajectory tracking maneuvers. The AAAP-PFA was evaluated on the TruckSim simulation platform with a nonlinear vehicle model and compared with the OPCDM over three tests.

The simulation results confirmed that the proposed PFA with a positive posture compensation coefficient can improve the overall path-following performance of the tractor-semitrailer, and the proposed PFA with a negative posture compensation coefficient can improve the lateral stability performance. Moreover, its adaptive ability to trade-off lateral stability and path-following performance, as well as strong steering control robustness, was demonstrated. Furthermore, a method to eliminate roll jitter and sudden yaw response need to be thoroughly investigated.

## Figures and Tables

**Figure 1 sensors-22-05163-f001:**
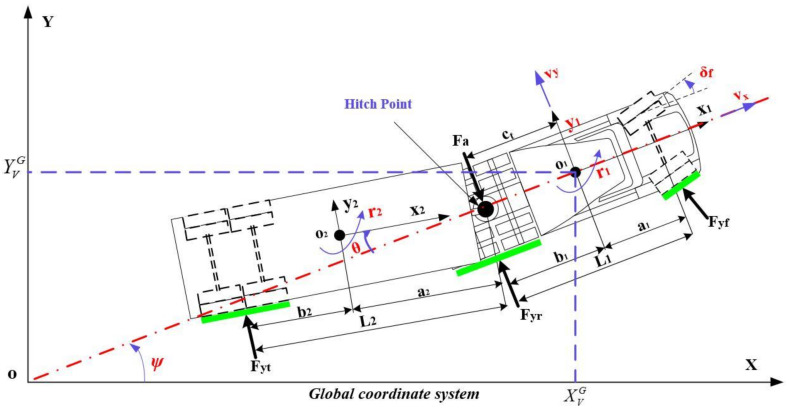
Three DOF linear yaw-model of tractor-semitrailer.

**Figure 2 sensors-22-05163-f002:**
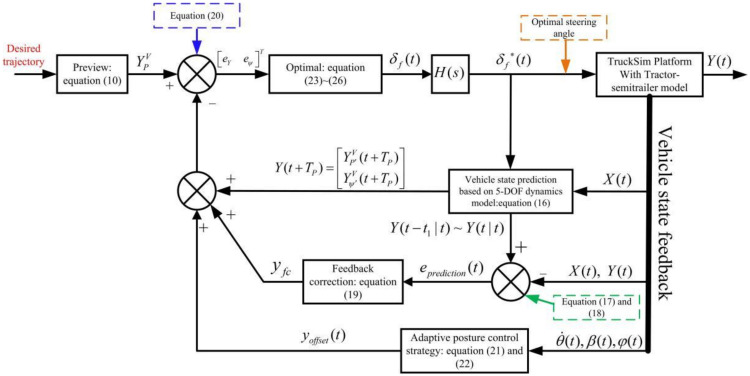
Overall block diagram of AAAP-PFA.

**Figure 3 sensors-22-05163-f003:**
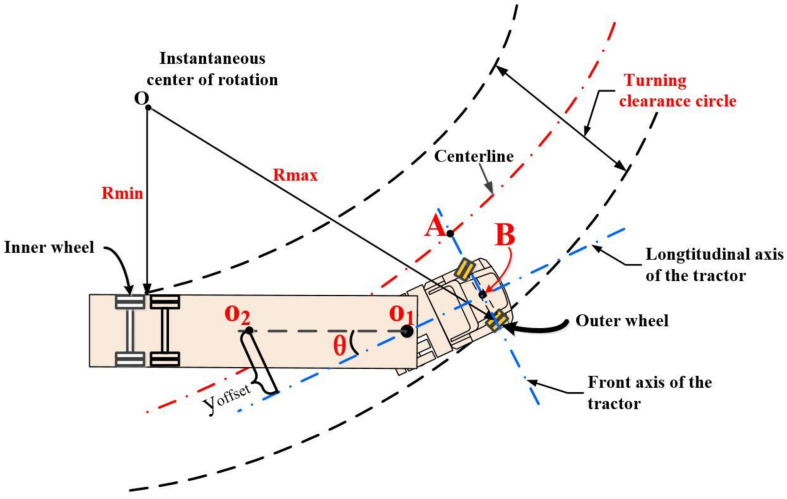
Steering kinematics model of a semitrailer.

**Figure 4 sensors-22-05163-f004:**
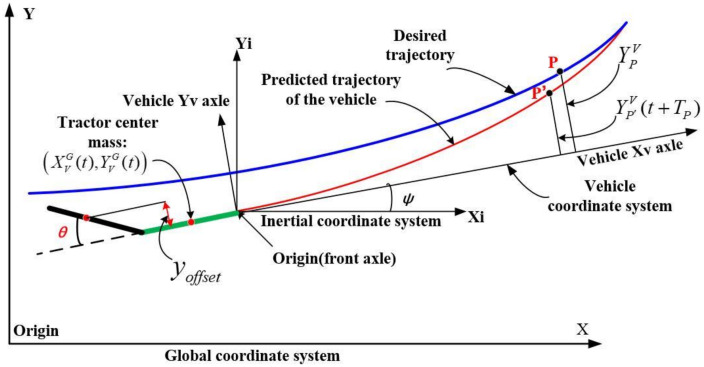
Preview principle of AAAP-PFA.

**Figure 5 sensors-22-05163-f005:**
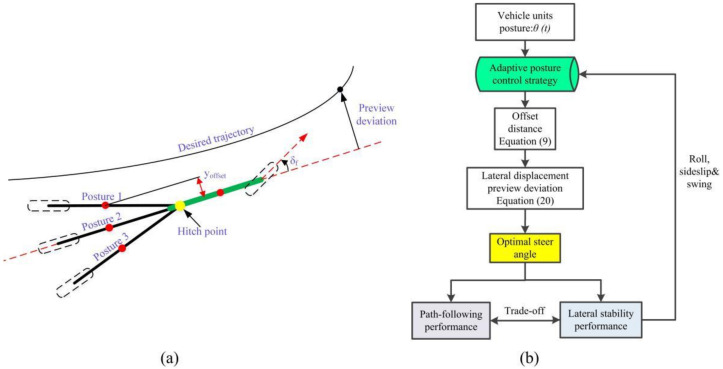
Principle of posture control. (**a**) Three postures. (**b**) Posture control flow.

**Figure 6 sensors-22-05163-f006:**
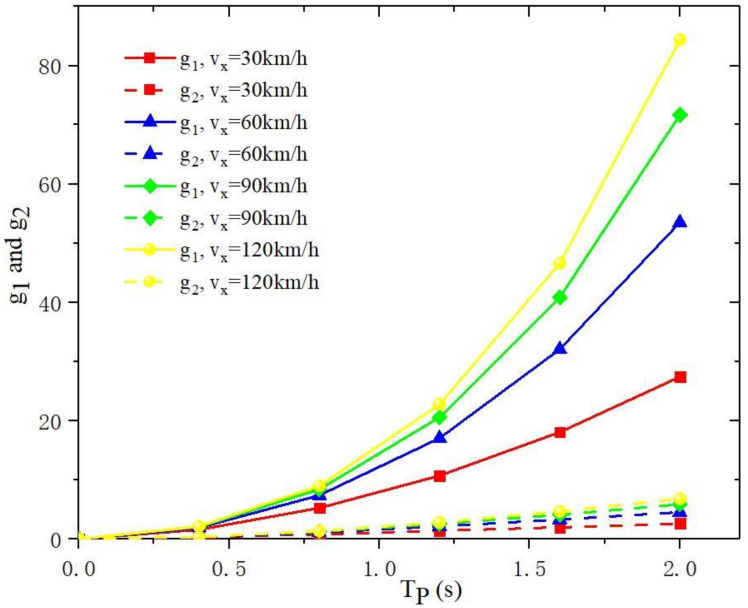
The values of *g*_1_ and *g*_2_ under different preview times *T_P_*.

**Figure 7 sensors-22-05163-f007:**
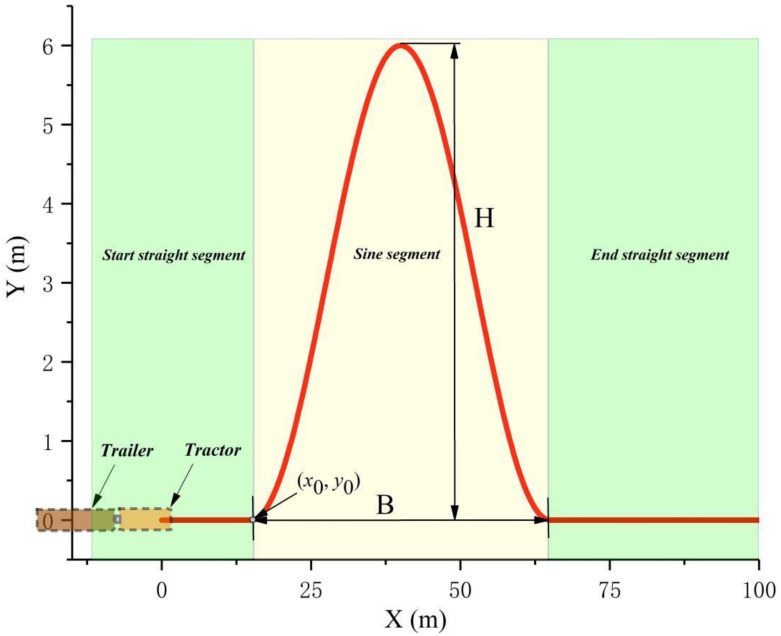
Sine trajectory.

**Figure 8 sensors-22-05163-f008:**
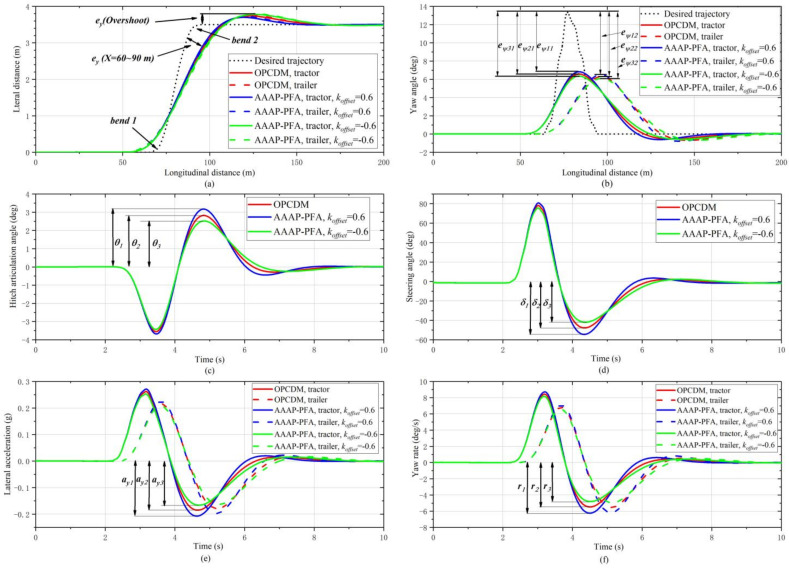
Simulation results of test 1 at a speed of 80 km/h. (**a**) Trajectory. (**b**) Yaw angle. (**c**) Hitch articulation angle. (**d**) Steering angle. (**e**) Lateral acceleration. (**f**) Yaw rate.

**Figure 9 sensors-22-05163-f009:**
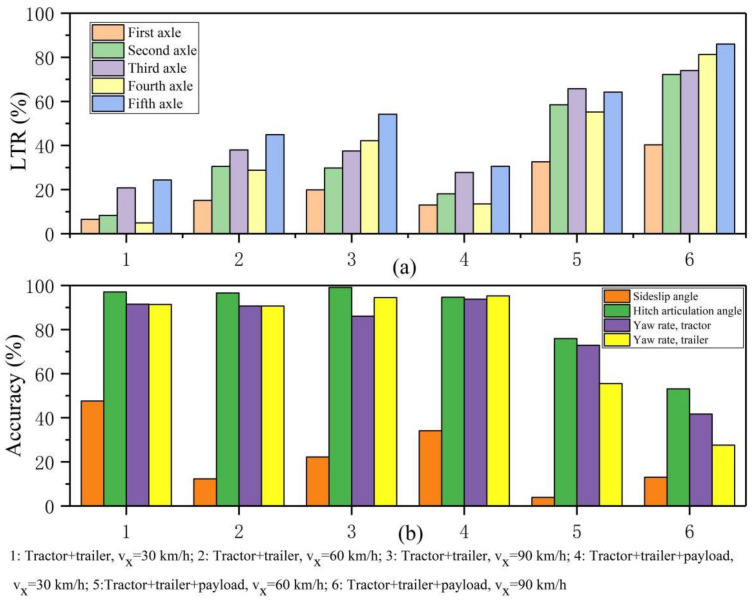
Accuracy of steady-state value prediction under different vehicle speeds and load conditions. (**a**) LTR. (**b**) Accuracy.

**Figure 10 sensors-22-05163-f010:**
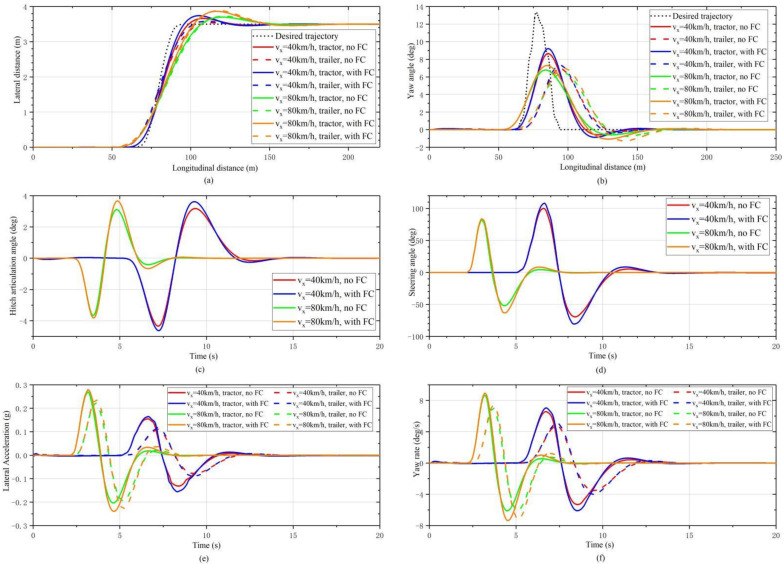
Simulation results of AAAP-PFA with and without FC. (**a**) Trajectory. (**b**) Yaw angle. (**c**) Hitch articulation angle. (**d**) Steering angle. (**e**) Lateral acceleration. (**f**) Yaw rate.

**Figure 11 sensors-22-05163-f011:**
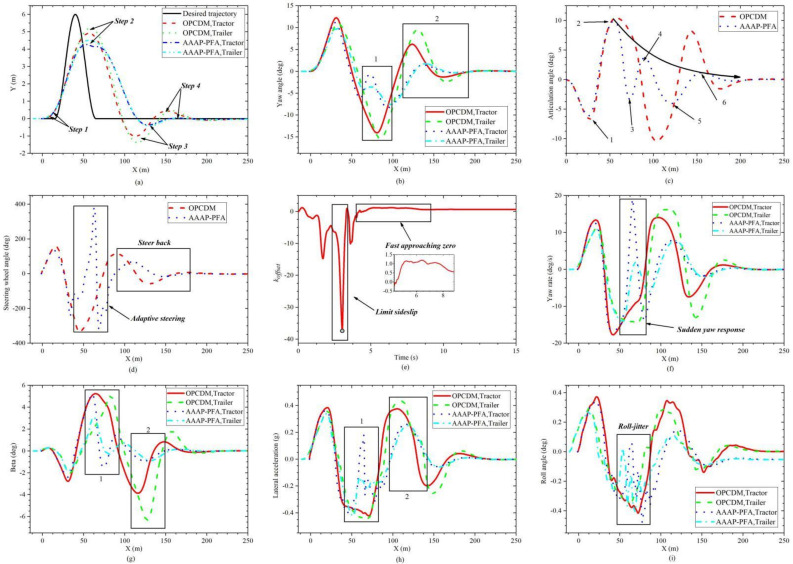
Comparison results of OPCDM and AAAP-PFA under low road adhesion conditions. (**a**) Trajectory. (**b**) Yaw angle. (**c**) Articulation angle. (**d**) Steering angle. (**e**) Koffset. (**f**) Yaw rate. (**g**) Beta. (**h**) Lateral acceleration. (**i**) Roll angle.

**Table 1 sensors-22-05163-t001:** Vehicle dynamics parameters.

Symbol	Description	Value
*m*_1_, *m*_2_	Whole mass of the tractor and trailer (kg)	8450, 7255
*a*_1_, *b*_1_	Longitudinal distance from the front axle and the rear axle to the CG of the tractor (m)	2.081, 3.554
*a*_2_, *b*_2_	Longitudinal distance from the hitch point and the rear axle to CG of the trailer (m)	6.365, 3.855
*c_t_*	Longitudinal distance from the hitch point to the CG of the tractor (m)	3.554
*L*_1_, *L*_2_	The wheelbase of the tractor and trailer (m)	5.635, 10.220
*C_yf_*, *C_yr_*, *C_yt_*	Combined cornering stiffness of the tires of the front axle and rear axle of the tractor and the rear axle of the tractor (N/rad)	135,010, 183,340, 152,450
*I*_Z1_, *I*_Z2_	Yaw moment of inertia of the tractor and trailer (kg·m^2^)	46,599, 206,680

**Table 2 sensors-22-05163-t002:** Effect of *k_offset_* on the preview deviation.

Poseture	*k_offset_* > 0	*k_offset_* = 0	*k_offset_* < 0
**1**	−	no effect	+
**2**	no effect	no effect	no effect
**3**	+	no effect	−

**Table 3 sensors-22-05163-t003:** Laws presented by the simulation results at bend 2.

Performance	Laws
Tracking deviation	*e_y_*_11_ < *e_y_*_21_ < *e_y_*_31_, *e_y_*_12_ < *e_y_*_22_ < *e_y_*_32_, *e_ψ_*_11_ < *e_ψ_*_21_ < *e_ψ_*_31_, *e_ψ_*_12_ < *e_ψ_*_22_ < *e_ψ_*_32_
Steering input	*δ*_1_ > *δ*_2_ > *δ*_3_
Lateral Stability Variables	*θ*_1_ > *θ*_2_ > *θ*_3_, *a_y_*_11_ > *a_y_*_21_ > *a_y_*_31_, *a_y_*_12_ > *a_y_*_22_ > *a_y_*_32_, *r*_11_ > *r*_21_ > *r*_31_, *r*_12_ > *r*_22_ > *r*_32_

## Data Availability

Not applicable.

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
