# Peer review of "Adaptive Articulation Angle Preview-Based Path-Following Algorithm for Tractor-Semitrailer Using Optimal Control"

_sensors, 2022, doi:10.3390/s22145163_

Round 1

Reviewer 1 Report

For the tractor-semitrailer path following problem, this paper proposes an adaptive Articulation angle preview-based path following algorithm. The algorithm adopts a feedback control structure, predicts the vehicle path from the linear prediction model, calculates the path deviation from the angle prediction for feedback correction control, and compares it with other related algorithms by simulation. There are several problems:

1. Why choose linear model as prediction model instead of nonlinear model as prediction path;

2. The model state equation (equation 8) is not clearly expressed, and the meaning of state X and output y is not given;

3. Since the model state equation (equation 8) is not clearly described, the overall control structure diagram (Figure 2) is not clearly described; In Figure 2, there are three feedback channels. The output of channel 1 is consistent with the input quantity (prediction y), while the outputs of the remaining two channels are feedback correction quantity yfc and adaptive attitude output yoffset respectively. How these three feedback outputs are combined into a feedback outputs is not clearly given in the article, which is inappropriate.

Conclusion: I suggest that this paper in this version cannot be published, which needs a major revision.

Author Response

Comment 1: Why choose linear model as prediction model instead of nonlinear model as prediction path;

Answer 1: Although linear tires can reduce the accuracy of the prediction model at high speeds, they can improve the calculation speed and reduce the complexity. In section 3.2, in order to solve the prediction mismatch caused by the linear model, a feedback correction method is proposed.Therefore, we adopt a linear model to predict vehicle trajectories and states.

Comment 2: The model state equation (equation 8) is not clearly expressed, and the meaning of state X and output y is not given;

Answer 2: We appreciate it very much for this good suggestion, and we have done it according to your ideas. The meaning of state X and output Y is given at the end of the section 2.

Comment 3: Since the model state equation (equation 8) is not clearly described, the overall control structure diagram (Figure 2) is not clearly described; In Figure 2, there are three feedback channels. The output of channel 1 is consistent with the input quantity (prediction y), while the outputs of the remaining two channels are feedback correction quantity yfc and adaptive attitude output yoffset respectively. How these three feedback outputs are combined into a feedback outputs is not clearly given in the article, which is inappropriate.

Answer 3: We have revised Figure 2 based on your comments. Readers can clearly understand how these three feedback outputs are combined into a feedback outputs according to the revised Figure 2. If there are other inappropriate expressions, please do not hesitate to raise them so that we can better improve this manuscript.

The above content is our revision and answer to your comments. For more detailed revisions, please see the attachment. Also, thank you for your valuable comments on our manuscript.

Reviewer 2 Report

The paper presents a finite-horizon optimal control algorithm for trajectory tracking of an articulated vehicle with feedback correction to compensate for model mismatch and adaptive posture correction. The control algorithm is suitable for the particular challenges posed by the problem of following a trajectory with an articulated vehicle (tractor or semitrailer) with minor uncertainties in the model. The numerical simulations are extensive and demonstrate the usefulness of the algorithm. However, the language needs some improvement (as an example in the second sentence of the abstract "Since ignore the motion of the trailer" is grammatically incorrect) and there are some open questions:

-This paper deals with steering control of an articulated vehicle to perform trajectory tracking. However, since the desired trajectory is parametrised by time and there are strict time requirements, it is apparent that some sort of speed control is required in order for the vehicle to not lag behind or remain ahead of its assigned trajectory. The authors should then clarify the issue of speed control.

-In section 2 the authors should specify what the output of the linear system is.

-On section 3.3 the authors should explain how to select an appropriate prediction horizon Tp, the estimation time t1 and the gains kfc1 and kfc2.

Author Response

Comment 1: The numerical simulations are extensive and demonstrate the usefulness of the algorithm. However, the language needs some improvement (as an example in the second sentence of the abstract "Since ignore the motion of the trailer" is grammatically incorrect) and there are some open questions:

Answer 1: We have improved the expression in the text based on your comments.

Comment 2: This paper deals with steering control of an articulated vehicle to perform trajectory tracking. However, since the desired trajectory is parametrised by time and there are strict time requirements, it is apparent that some sort of speed control is required in order for the vehicle to not lag behind or remain ahead of its assigned trajectory. The authors should then clarify the issue of speed control.

Answer 2: We have improved on the first assumption at the beginning of the section 2 based on your suggestion. This paper mainly focuses on the lateral stability of the vehicle body units and the tracking deviation during the trajectory tracking process. The new assumption is “forward speed is a constant value, and is maintained by a PID algorithm embedded in the software”.

Comment 3: In section 2 the authors should specify what the output of the linear system is.

Answer 3: We define the linear output at the end of the section 2 and clearly describe the logical relationship between the equations in Figure 2.

Comment 4: On section 3.3 the authors should explain how to select an appropriate prediction horizon Tp, the estimation time t1 and the gains kfc1 and kfc2.

Answer 4: We have explained the basis for the selection of parameters t1, kfc1 and kfc2 in section 3.2 based on your comments. The following is the specific expression:

(1) Considering the calculation efficiency and the accumulated error in the prediction process, after many simulation tests, when 0.02 TP ≤t1 and Δt ≤ 0.2 TP, a better control effect can be achieved.

(2) kfc1 and kfc2 need to be adaptively adjusted according to the accuracy of the prediction results.

(3) The selection basis of TP involves control stability and trajectory tracking deviation, which has been analyzed in Section 3.5.

The above content is our revision and answer to your comments. For more detailed revisions, please refer to the revised version of the manuscript. Also, thank you for your valuable comments on our manuscript.

Round 2

Reviewer 1 Report

 I suggest that this paper in this version can be published